# Different Effects of RNAi-Mediated Downregulation or Chemical Inhibition of NAMPT in an Isogenic IDH Mutant and Wild-Type Glioma Cell Model

**DOI:** 10.3390/ijms23105787

**Published:** 2022-05-21

**Authors:** Maximilian Clausing, Doreen William, Matthias Preussler, Julia Biedermann, Konrad Grützmann, Susan Richter, Frank Buchholz, Achim Temme, Evelin Schröck, Barbara Klink

**Affiliations:** 1Institute for Clinical Genetics, University Hospital Carl Gustav Carus, Technische Universität Dresden, ERN-GENTURIS, Hereditary Cancer Syndrome Center Dresden, 01307 Dresden, Germany; maximilian.clausing@uniklinikum-dresden.de (M.C.); doreen.william@uniklinikum-dresden.de (D.W.); matthias.preussler@uniklinikum-dresden.de (M.P.); julia.biedermann@uniklinikum-dresden.de (J.B.); evelin.schroeck@uniklinikum-dresden.de (E.S.); 2National Center for Tumor Diseases Partner Site Dresden (NCT/UCC), German Cancer Consortium (DKTK), 01307 Dresden, Germany; frank.buchholz@tu-dresden.de (F.B.); achim.temme@uniklinikum-dresden.de (A.T.); 3German Cancer Research Center (DKFZ), 69120 Heidelberg, Germany; konrad.gruetzmann@tu-dresden.de; 4Core Unit for Molecular Tumor Diagnostics (CMTD), National Center for Tumor Diseases (NCT/UCC), German Cancer Consortium (DKTK), 01307 Dresden, Germany; 5Institute for Clinical Chemistry and Laboratory Medicine, University Hospital Carl Gustav Carus, Technische Universität Dresden, 01307 Dresden, Germany; susan.richter@uniklinikum-dresden.de; 6Medical Systems Biology, Faculty of Medicine, University Hospital Carl Gustav Carus, Technische Universität Dresden, 01307 Dresden, Germany; 7Section Experimental Neurosurgery/Tumor Immunology, Department of Neurosurgery, University Hospital Carl Gustav Carus, Technische Universität Dresden, 01307 Dresden, Germany; 8National Center of Genetics (NCG), Laboratoire National de Santé (LNS), L-3555 Luxembourg, Luxembourg

**Keywords:** IDH1 mutation, glioma, redox household, nicotinamide phosphoribosyltransferase, NAD^+^ synthesis

## Abstract

The IDH1^R132H^ mutation in glioma results in the neoenzymatic function of IDH1, leading to the production of the oncometabolite 2-hydroxyglutarate (2-HG), alterations in energy metabolism and changes in the cellular redox household. Although shifts in the redox ratio NADPH/NADP^+^ were described, the consequences for the NAD^+^ synthesis pathways and potential therapeutic interventions were largely unexplored. Here, we describe the effects of heterozygous IDH1^R132H^ on the redox system in a CRISPR/Cas edited glioblastoma model and compare them with IDH1 wild-type (IDH1^wt^) cells. Besides an increase in 2-HG and decrease in NADPH, we observed an increase in NAD^+^ in IDH1^R132H^ glioblastoma cells. RT-qPCR analysis revealed the upregulation of the expression of the NAD^+^ synthesis enzyme nicotinamide phosphoribosyltransferase (NAMPT). Knockdown of NAMPT resulted in significantly reduced viability in IDH1^R132H^ glioblastoma cells. Given this dependence of IDH1^R132H^ cells on NAMPT expression, we explored the effects of the NAMPT inhibitors FK866, GMX1778 and GNE-617. Surprisingly, these agents were equally cytotoxic to IDH1^R132H^ and IDH1^wt^ cells. Altogether, our results indicate that targeting the NAD^+^ synthesis pathway is a promising therapeutic strategy in IDH mutant gliomas; however, the agent should be carefully considered since three small-molecule inhibitors of NAMPT tested in this study were not suitable for this purpose.

## 1. Introduction

Gliomas are the most prevalent histological type of primary malignant central nervous system tumors and one of the most malignant types of cancer according to their aggressive invasive potential [1,2]. Current therapies using combinations of surgery, radiotherapy and chemotherapy have limited success with 10-year overall survival rates of less than 1% for glioblastoma [3,4]. The non-specific nature of current treatments might be one factor. Therefore, the identification of new therapeutic strategies more specifically targeting tumor cells is of great interest.

Mutations of the key Krebs cycle enzyme isocitrate dehydrogenase (IDH) are considered to be vital for the genesis of low-grade gliomas and secondary glioblastomas [5], prompting the World Health Organization (WHO) to separate gliomas according to IDH mutation status into IDH mutant and IDH wild-type (IDH1^wt^) entities in the “2016 WHO Classification of Tumors of the Central Nervous System” [6]. The most frequent of these mutations is IDH1^R132H^—the point mutation of arginine to histidine at residue 132, accounting for more than 80% of IDH mutations [7]. It results in a neoenzymatic function of IDH1, leading to the near-complete elimination of the oxidation of isocitrate to α-ketoglutarate (α-KG) catalyzed by IDH1^wt^. Instead, IDH1^R132H^ leads to the NADPH-consuming production of 2-hydroxyglutarate (2-HG) via the reduction of α-KG [8,9]. 2-HG is accepted as an oncometabolite, with many studies having been focused on oncogenic effects through the inhibition of α-KG dependent dioxygenases [10,11], such as hypoxia-inducible factor (HIF) hydroxylases and methylcytosine dioxygenase TET2, resulting in altered HIF activity and CpG island-methylator phenotype (G-CIMP) [12,13].

Besides these 2-HG-mediated oncogenic effects, other metabolic changes were described in IDH1 mutated gliomas. These alterations include decreases in glutathione metabolites [14], activation of glutaminolysis [15], aberrations in lipid metabolism and reduced glucose turnover [16]. Moreover, IDH1 mutations were shown to alter redox metabolism in glioma cells. Reported drops in NADPH levels may be a direct result of the NADPH-consuming reaction catalyzed by mutated IDH1, whereas wild-type IDH1 was identified as the main source of cytosolic NADPH in glia cells and glioblastoma [17,18]. We and others recently also described a significant decrease in NAD^+^ levels in glioma cells with IDH1^R132H^, leading to the hypothesis of NAPDH restoration via the phosphorylation of NAD^+^ and the identification of NAD^+^ synthesis inhibition as a possible treatment [17,19]. Intriguingly, we found that NAD^+^ levels are not altered in IDH1 mutated astrocytes [17], indicating successful compensatory mechanisms in those cells and, therefore, potentially limiting the vulnerability of cancer cells to NAD^+^ synthesis inhibition.

Nicotinamide phosphoribosyltransferase (NAMPT) catalyzes the rate-limiting conversion of nicotinamide to nicotinamide mononucleotide (NMN) as part of the NAD^+^ salvage pathway [20]. Of the various NAD^+^ synthesis enzymes, we found NAMPT to be the only one ubiquitously expressed among different patient-derived glioma cell lines, making it a natural target in those cells [17]. NAMPT was identified as a promising anticancer target due to the absence of other NAD+ synthesis enzymes in several tumors (e.g., prostate carcinoma, sarcomas, neuroblastomas and glioblastomas) [21,22,23], its association with worse prognosis in glioblastomas [24] and its overexpression in several types of tumor cells including gliomas [25]. NAMPT small-molecule inhibitors were shown to induce cytotoxicity through NAD^+^ depletion in a wide range of tumor models in vitro and in vivo (e.g., colorectal carcinoma, acute myeloid leukemia and glioblastoma) [26].

This study focused on redox metabolism as a treatment option in IDH1 mutated gliomas. We created a novel cell model by inducing the heterozygous IDH1^R132H^ point mutation at the target genomic sequence into a primary patient-derived glioblastoma cell line using CRISPR/Cas9. In light of the difficulties of culturing IDH1 mutated patient-derived glioma cells in vitro [27,28] and the lack of patient-derived IDH mutant in vitro and in vivo models [29,30], our approach provides a suitable alternative to investigate IDH1^R132H^-dependent alterations in tumor cell metabolism. Our endogenous IDH1^R132H^ cell model showed 100-fold elevated 2-HG levels, confirming the functionality of the mutated IDH1 enzyme and making it comparable to patient samples [8]. In this work, we explored the effects of IDH1^R132H^ on NAD^+^ metabolism and the therapeutic potential of NAMPT knockdown and NAMPT small-molecule inhibitors in IDH1^wt^ and IDH1^R132H^ glioma cells.

## 2. Results

### 2.1. Alterations in NAD^+^ and NADP^+^ Metabolism in IDH1^R132H^ Cells Could Be Directly Attributed to the Neoenzymatic Function of IDH1^R132H^

We successfully introduced the IDH1 c.395G > A point mutation heterozygously in a primary patient-derived glioblastoma cell line (HT7606 [31]) using CRISPR/Cas9. Three IDH1^R132H^ clones, as well as three IDH1^wt^ clones as controls, were created. The mutation was stable in all three IDH1^R132H^ clones in a long-time culture for at least 50 passages (Appendix A). We confirmed the IDH1 and IDH1^R132H^ protein expression (Appendix A) and the functionality of the IDH1^R132H^ enzyme. The 2-HG level was approximately 100-fold elevated in IDH1^R132H^ mutated cells compared with the IDH1^wt^ cells. Treatment with the mutant IDH1 inhibitor AGI-5198 normalized 2-HG levels in IDH1^R132H^ cells, confirming the functionality of the inhibitor and suitability of our cell line model (Appendix A). The results for the CRISPR/Cas-edited cell lines are comparable with patient samples [8] and verified the functionality of the mutated enzyme.

Intracellular NAD^+^ and NADPH^+^ levels were previously reported to differ in gliomas in a manner that was dependent on the IDH1 status [16,17,19,32]. To investigate alterations in the NADP^+^/H and NAD^+^/H metabolism in our isogenic IDH1^R132H^ cell models, we measured the NADPH/NADP^+^ and NADH/NAD^+^ ratios, as well as the NADP^+^, NADPH, NAD^+^ and NADH levels. The NADPH/NADP^+^ ratios were significantly lower in IDH1^R132H^ cells compared with IDH1^wt^ cells due to a significant decrease in NADPH levels (Figure 1A). Furthermore, the NADH/NAD^+^ ratios were also significantly lower in IDH1^R132H^ cells (Figure 1B). This effect was caused by increases in the NAD^+^ levels in the mutated cells, indicating an upregulation of NAD^+^ synthesis in IDH1^R132H^ cells. To confirm that the observed changes in redox metabolism were directly linked to the neo-enzymatic activity of IDH1^R132H^, the cells were treated with the specific mutant IDH1 inhibitor AGI-5198. Rescue of the NADPH level and NADPH/NADP^+^ ratio upon treatment with AGI-5198 in IDH1 mutant cells confirmed this to be a direct effect due to the increased consumption of NADPH by the IDH1^R132H^ enzyme (Figure 1C). Notably, treatment with AGI-5198 also normalized the NAD^+^ levels in IDH1^R132H^ mutated cells (Figure 1D). Therefore, the observed changes in redox metabolism could be directly attributed to the functionality of IDH1^R132H^. We hypothesize that IDH1^R132H^ leads to upregulation of NAD^+^ synthesis via NAMPT to restore consumed NADPH via the phosphorylation of NAD^+^.

### 2.2. IDH1^R132H^ Altered NAMPT Expression

We previously identified NAMPT as the only NAD^+^ synthesis enzyme that is ubiquitously expressed in IDH mutant and IDH wild-type gliomas and that NAMPT protein expression is lower in IDH mutant gliomas [17]. We, therefore, analyzed the expression of NAMPT on mRNA and protein levels in our IDH1^wt^ and IDH1^R132H^ cells. NAMPT mRNA levels of IDH1^wt^ cells were comparable to those in normal brain tissue (Figure 2A). IDH1^R132H^ cells exhibited approximately three-fold higher NAMPT mRNA expression, supporting our hypothesis of upregulated NAD^+^ synthesis. However, analysis of the NAMPT protein levels revealed similar expressions in IDH1^R132H^ and IDH1^wt^ cells (Figure 2B). The discrepancy between changes in NAMPT mRNA and protein expression suggests that posttranslational regulation of NAMPT expression differs between IDH1^R132H^ and IDH1^wt^ cells.

### 2.3. Knockdown of NAMPT and other NAD^+^ Synthesis Enzymes Selectively Reduced the Viability of IDH1^R132H^ Cells

To investigate the effects of NAMPT gene knockdown as a potential target for the treatment of IDH mutant glioma, IDH1^wt^ and IDH1^R132H^ cells were transfected using two different esiRNAs targeting different sequences of NAMPT mRNA (esiNAMPT-A and esiNAMPT-B) to confirm the esiRNA specificity. The RT-qPCR results revealed knockdown efficiencies of approximately 95% for esiNAMPT-A and 75% for esiNAMPT-B 72 h post-transfection (Figure 3A,C). NAMPT protein levels were considerably lower 48 h post-transfection of esiNAMPT-A in IDH1^wt^ and IDH1^R132H^ compared with the mock control treatment cells and no longer detectable 72 h after transfection (Figure 3B). Likewise, NAMPT protein levels were not detectable 72 h post-transfection of esiNAMPT-B (Figure 3D).

Forty-eight hours after the NAMPT knockdown, the NAD(P)H-dependent WST-1-reducing capability was significantly diminished in both the IDH1^wt^ and IDH1^R132H^ cells (Figure 4A), indicating a decrease in the reducing agents due to a lack of functional NAMPT. NAD^+^ levels were higher in untreated IDH1^R132H^ cells compared with IDH1^wt^ but reached similar levels after NAMPT knockdown (Figure 4B). NADP^+^ levels were unaltered by NAMPT knockdown. NADPH levels, on the other hand, dropped in both cell lines, reducing the NADPH levels in IDH1^R132H^ cells even further to about one-third of that of IDH1^wt^. NADH levels dropped in IDH1^R132H^ cells after NAMPT knockdown by nearly 80%, but not in IDH1^wt^. NAMPT knockdown resulted in a decrease in cell viability of 45% for esiNAMPT-A and 59% for esiNAMPT-B in IDH1^R132H^ cells, as determined by fluorescent-based quantification of DNA content (Figure 4C). Despite the observed change in WST-1-reducing capacity, cell viability was not affected in IDH1^wt^ cells. These results indicated an enhanced dependence of IDH1^R132H^ cells on NAD^+^ synthesis compared with IDH1^wt^ cells.

To confirm the selective dependence of IDH1^R132H^ cells on NAD^+^ synthesis, we analyzed the effects of NAMPT knockdown using live cell counting and extended our study to other enzymes involved in NAD^+^ synthesis pathways (Figure 4D). We previously analyzed the expression levels of the rate-limiting enzymes of the four NAD^+^ synthesis pathways in glioma cells and astrocytes and found the two salvage pathways via NMRK1 and NAMPT to be the sole contributors to NAD^+^ synthesis [17]. Thus, we performed a knockdown of NMRK1, the rate-limiting enzyme of the second salvage pathway, as well as a knockdown of NMNAT1, which is the last step in NAD synthesis in all four NAD^+^ synthesis pathways, in our cell models. We confirmed the successful knockdown after esiRNA treatment using Western blotting (Appendix A). Live cell counting with the Operetta High-Content Imaging System confirmed the selective reduction in cell viability in IDH1^R132H^ cells after the knockdown of NAMPT with esiNAMPT-A and esiNAMPT-B (Figure 4E). The knockdown of NMRK1 or NMNAT1 resulted in a similar selective effect on IDH1^R132H^ cells.

To investigate whether NAD^+^ synthesis might be upregulated in IDH1^R132H^ cells to restore consumed NADPH via phosphorylation of NAD^+^, we performed an esiRNA-mediated knockdown of NADK, which is the enzyme catalyzing the ATP-dependent phosphorylation of NAD^+^ to NADP^+^. In line with our hypothesis, NADK knockdown reproduced the effects of NAMPT and NMRK1 knockdowns (Figure 4E). Taken together, these data showed an increased dependence of IDH1^R132H^ glioma cells on NAD^+^ synthesis and phosphorylation, suggesting that the investigated pathways might be promising therapeutic targets for gliomas with an IDH1^R132H^ mutation.

### 2.4. Effects of NAMPT Small-Molecule Inhibitors on Cell Viability Were Independent of IDH1 Status

Several NAMPT small-molecule inhibitors were recently described as potential anticancer agents [26]. To test whether pharmacological NAMPT inhibition resulted in similar cytotoxicity in IDH1^R132H^ cells to the esiRNA-mediated NAMPT knockdown, we exposed our cells to the chemically distinct, specific NAMPT inhibitors FK866, GMX1778 and GNE-617. Each inhibitor reduced the metabolic activity—quantified via the NAD(P)H-dependent WST-1-reduction rate—after 48 h of treatment in a concentration-dependent manner (Figure 5A). The IC50 values were determined to be 36.8 nM, 19.9 nM and 27.9 nM for FK866, GMX1778 and GNE-617, respectively. The viability of our cell models was unaffected after 48 h treatment with NAMPT inhibitors at concentrations of 100 nM, which corresponded to the maximal inhibitory concentrations measured using a WST-1 assay (Figure 5B). After 72 h, cytotoxic effects were observed in both IDH1^wt^ and IDH1^R132H^ cells, with an additional decrease in viability after 96 h of incubation. These effects were confirmed in another cell viability assay based on fluorescence quantification (Figure 5C). After 96 h of NAMPT inhibitor treatment, all investigated cells showed significantly reduced cell viability. In fact, the cytotoxicity of NAMPT inhibitors was even more severe in IDH1^wt^ cells, contrasting with the higher sensitivity of IDH1^R132H^ cells to esiRNA-mediated NAMPT knockdown. Lower inhibitor concentrations of 25 nM, approximately corresponding to the IC50, resulted in similar effects; the cell viability was more reduced in IDH1^wt^ compared to IDH1^R132H^ cells. NAMPT inhibitor concentrations of 1 nM were not sufficient to impair the cell viability, whereas cytotoxicity at very high concentrations (10 µM) did not substantially differ from the described results at 100 nM (Appendix A). In summary, the IDH1^R132H^ selective effect of NAMPT-knockdown was not reproducible using the NAMPT small-molecule inhibitors FK866, GMX1778 and GNE-617.

### 2.5. Combinatorial esiNAMPT and GMX1778 Treatment Indicated Unspecific Effects of Small Molecule NAMPT Inhibitors

To further investigate the effects of NAMPT small-molecule inhibitors in our cell models, we determined the NAD(H) and NADP(H) levels after 48 h of treatment with 25 nM (near IC50) GMX1778 (Figure 6A). Independently of the IDH1 status, the GMX1778 treatment reduced NAD^+^ and NADH to unmeasurable levels, explaining the unselective cytotoxicity in our cell models. Accordingly, the NADPH levels were significantly and strongly reduced in both the IDH mutant and wild-type cells. Since different NAD^+^ and NADP^+^ synthesis pathways exist, complete loss of NAD(H) after treatment with GMX1778 indicated that inhibition of NAD^+^ synthesis might be non-selective and not NAMPT specific.

We, therefore, explored the effects of treatment with GMX1778 on cell viability after the knockdown of NAMPT using esiNAMPT-A. Silencing with esiNAMPT-A led to a 95% effective NAMPT knockdown at 48 h and 72 h after the treatment (Figure 3B). For combinatory esiNAMPT-A and GMX1778 treatment, we applied GMX1778 48 h after the esiRNA transfection (Figure 6B). If GMX1778 specifically inhibited NAMPT, we would not expect an additional effect of GMX1778 in NAMPT knockdown cells. The GMX1778 treatment reduced viability in IDH1^wt^ and IDH1^R132H^ cells after 96 h (Figure 6B). In this experimental setup, IDH1^R132H^ cells showed a higher reduction in cell viability than IDH1^wt^. Intriguingly, the GMX1778 treatment caused the same non-selective effect after the knockdown of NAMPT in both wild-type and IDH mutant cells (Figure 6B). The observation that GMX1778 treatment also affected NAMPT knockdown cells strongly indicated nonspecific effects besides NAMPT inhibition.

## 3. Discussion

IDH1 mutations were previously shown to alter redox metabolism in glioma cells. Besides differences in NADPH levels that may be directly attributed to the neoenzymatic function of mutated IDH1, changes in NAD^+^ levels were also described [17,19]. Here, we confirmed the alteration of NADPH and NADH levels in a newly created IDH1^R132H^ mutated glioma cell model. The silencing of various enzymes involved in NAD^+^ synthesis and phosphorylation revealed a striking susceptibility of IDH1^R132H^ cells to those treatments compared with IDH1^wt^ cells, confirming increased dependence on NADPH synthesis. In contrast, small-molecule inhibitors of NAMPT—a central NAD^+^ synthesis enzyme—could not reproduce the distinction between IDH1^R132H^ cells and IDH1^wt^ cells, indicating different responses of the cells to NAMPT depletion on mRNA and pharmacological inhibition.

We found that the NADPH/NADP^+^ ratios were significantly decreased in IDH1^R132H^ cells compared with IDH1^wt^ cells, which was attributable to lower NADPH levels in those cells. This observation was in line with previous findings of decreased NADP^+^-dependent IDH activity [18,33] and confirmed observations from us and others of decreased NADPH/NADP^+^ ratios in stably transduced IDH1^R132H^ glioma cells [7,16,17]. As NADP^+^-dependent wild-type IDH is the main generator of NADPH in glioblastoma [18], the loss of NADPH was most likely a direct consequence of the NADPH-consuming neoenzymatic activity of mutated IDH1 [8]. Intriguingly, the NADH/NAD^+^ ratios were also decreased in our IDH1^R132H^ cells due to increased NAD^+^ levels. We hypothesize that the higher NAD^+^ levels are the result of the compensatory upregulation of NAD^+^ synthesis as a reaction to the above described NADPH loss. The NADPH pool, which is instrumental as an antioxidant for ROS scavenging, can be replenished via phosphorylation of NAD^+^ via NADK to NADP^+^ and a subsequent reduction to NADPH.

Treatment with AGI-5198 [34], a selective inhibitor of mutant IDH1, normalized both the NADPH/NADP^+^ and NADH/NAD^+^ ratios in IDH1^R132H^ to the level of IDH1^wt^ cells. Previous studies showed increased NADPH levels in IDH1^R132H^ cells after AGI-5198 treatment [35]; however, NADH and NAD^+^ levels were not investigated. The normalization of the NADPH/NADP^+^ ratio could be explained by the restoration of NADPH production capacity of IDH1^R132H^ cells after inhibition of IDH1^R132H^ [33]. Consequently, the proposed compensatory maintenance of high NAD^+^ levels would be expendable. Here, we show for the first time that IDH1^R132H^ inhibition indeed also normalized NADH/NAD^+^ ratios, which underlined the linkage of NAD^+^ and NADP^+^ homeostasis, hence supporting our hypothesis of compensatory upregulation of NAD^+^ synthesis in IDH1^R132H^ cells.

There are four known NAD^+^ synthesis pathways in mammalian cells starting from the substrates nicotinamide, nicotinamide riboside, nicotinic acid and tryptophan [36]. Previous analysis of key enzymes of those pathways identified NAMPT as the only one expressed in all investigated glioma cell lines and patient-derived glioma cell models [17]. Along with the reported overexpression of NAMPT in glioma cells, as well as its proposed association with oncogenic effects and poor prognosis in glioma [37,38], this made NAMPT an intriguing target. We found that IDH1^R132H^ leads to the upregulation of NAMPT expression on mRNA level in our glioma cell model, in line with NAD^+^ synthesis upregulation in those cells. However, the NAMPT protein levels were similar in the IDH1^R132H^ and IDH1^wt^ cells. These results confirmed our previous findings of lower NAMPT protein levels in IDH1^R132H^ cells compared with IDH1^wt^ cells in vitro and in vivo, as well as a discrepancy between the NAMPT mRNA and protein levels [17]. This indicated distinct post-transcriptional regulation of NAMPT expression in IDH1^R132H^ and IDH^wt^ cells. Apart from the well-described effects of IDH1^R132H^-produced 2-HG on DNA methylation, the oncometabolite also seems to impact the regulation of mRNA translation [39]. However, those mechanisms and their clinical implications remain to be investigated.

The knockdown of NAMPT expression led to reduced metabolic activity (shown by the WST1 level) independently of the IDH1 status, but only in IDH1^R132H^ cells it also resulted in reduced viability. The reduced metabolic activity in both IDH1^R132H^ and IDH1^wt^ cells could be explained as a direct consequence of the deficiency of NAD^+^ and its derivatives following the loss of the NAD^+^ salvage pathway from nicotinamide. The observed IDH1^R132H^ cell-specific reduction in cell viability after the NAMPT knockdown therefore indicated an enhanced dependence on NAD^+^ synthesis in IDH1^R132H^ cells compared with IDH1^wt^. This phenomenon might be explained by lower basal NADPH levels in IDH1^R132H^ cells, rendering them more vulnerable to the depletion of NAD^+^, which, according to our hypothesis, is used for NADPH replenishment. Unexpectedly, NAMPT knockdown only slightly reduced the NAD^+^ level in IDH1^R132H^ cells directly but instead reduced the NADH level in IDH1^R132H^ cells significantly. According to our observations, the IDH1^R132H^ cells might unavoidably replenish the NADPH level via phosphorylation of NADH by NADK2. Therefore, a lack in NAD^+^ regeneration could result in diminished NADH level marking NADK2 as another therapy target.

NADPH synthesis from NAD^+^ requires the enzyme NADK for phosphorylation of NAD^+^ to NADP^+^. If NAD^+^ is indeed the precursor for the regeneration of scarce NADPH in IDH1^R132H^ cells, one would expect an increased dependence of those cells, not only on NAMPT but also on NADK, as well as other NAD^+^ synthesis enzymes. We previously found that NAD^+^ restoration in glioblastoma cells utilizes the salvage pathways via NAMPT and NMRK1, while nicotinic acid phosphoribosyltransferase (NAPRT) and de novo synthesis from tryptophan via quinolinic acid phosphoribosyltransferase (QPRT) are not involved [17]. Therefore, to examine the dependence on NAD^+^ synthesis and phosphorylation, we expanded our investigation to NADK, NMRK1 and NMNAT1—the enzyme downstream of NAMPT, NMRK1, NAPRT and QPRT—in addition to NAMPT. The selective vulnerability of IDH1^R132H^ cells to knockdown of NAMPT and NMRK1 indicated that these cells used both pathways to synthesize NAD^+^, making both enzymes possible targets for the selective treatment of IDH1 mutated tumor cells. However, NMRK1 expression varied substantially between different glioma cells [17], thus limiting the clinical practicality of NMRK1-inhibiting approaches. The cytotoxicity of NMNAT1 knockdown was not only seen in IDH1^R132H^ cells but also, to a smaller extent, in IDH1^wt^ cells. This observation, as well as the greater effect of NMNAT1 knockdown on IDH1^R132H^ cells compared to NAMPT or NMRK1 knockdown, seems plausible considering the involvement of NMNAT1 in all available NAD^+^ synthesis pathways. IDH1^wt^ cells seem to be able to compensate for the loss of only one pathway, making NMNAT1 a less attractive target for selective therapeutic approaches. Furthermore, we found a selective susceptibility of IDH1^R132H^ cells to NADK knockdown, confirming the increased dependence of those cells on NADPH regeneration via NAD^+^ phosphorylation.

As a possible clinical approach to make use of the increased dependence of IDH1^R132H^ cells on NAD^+^ synthesis and regeneration, we examined small molecular NAMPT inhibitors FK866, GMX1778 and GNE-617, which were shown to induce cell death in a variety of tumor cells and have, in part, completed phase I trials [40]. The decrease in metabolic activity we found after treatment with these agents complies with our data after esiRNA-mediated NAMPT inhibition, as well as reports of decreased levels of NAD^+^, NADH, NADP^+^ and NADPH in glioma cells with and without IDH1 mutation [19,41,42,43,44]. Other studies revealed that the reduction in metabolic activity preceded the cytotoxic effects of NAMPT inhibitors by a few hours [22,45,46]. Accordingly, we found that cytotoxic effects did not arise until 24 h after metabolic impairment in our cells. Contrary to the NAMPT esiRNA treatment, those effects were not limited to IDH1^R132H^ cells but occurred to a similar extent in IDH1^wt^ cells. A possible explanation could be the off-target effects of the NAMPT inhibitors, yet several studies demonstrated their specific activity toward NAMPT inhibition [22,45]. Using a combinatorial treatment of NAMPT knockdown and subsequent GMX1778 addition, we confirmed the cytotoxic effect of this inhibitor in the absence of NAMPT expression, confirming that GMX1778 induced off-target mediated cytotoxicity. Hasmann et al. concluded that FK866 has very low nonspecific cytotoxicity by revealing a lack of acceleration of apoptosis induction when using 100-fold IC50 concentrations of FK866. Furthermore, they ascribed the NAD^+^ depleting effect of FK866 to the inhibition of NAMPT by showing that NAD^+^ synthesis from nicotinic acid was not impaired by FK866. Subsequently, a reduction in NAMPT activity resulting from FK866 treatment was confirmed via measurement of the radioactive nicotinamide mononucleotide formed from the ^14^C-labeled substrate nicotinamide [45]. Watson et al. found NAD^+^ to be the metabolite that was most profoundly changed in cells exposed to GMX1778. They also showed that GMX1778 had no effect on the NAD^+^ synthesis from nicotinic acid and identified GMX1778 as an inhibitor of NAMPT by measuring NAMPT enzyme activity and the binding affinity of NAMPT for GMX1778 [22]. In conclusion, both studies demonstrated the specificity of FK866 or GMX17 toward NAMPT inhibition regarding the NAD^+^-depleting effects of those inhibitors. However, possible off-target effects concerning other metabolic pathways have not been investigated thoroughly.

The described discrepancy between NAMPT mRNA and protein expression might also contribute to the different responses to NAMPT knockdown on the mRNA level and NAMPT protein inhibition. Glioma cells overexpressing NAMPT were shown to be more sensitive to its inhibition [38]. Our NAMPT mRNA expression and esiRNA knockdown data revealed a similar correlation. However, NAMPT protein levels affected by possible post-transcriptional regulation of NAMPT expression in IDH1^R132H^ cells could result in low on-target effects of the tested inhibitors. This observation might prevent selective cellular impact and a call for concentrations of NAMPT inhibitors at which systemic cellular cytotoxicity predominates and the survival advantage of IDH1^wt^ cells is lost. Accordingly, low concentrations of NAMPT inhibitors did not induce cytotoxicity in any of the cells.

Contrasting with our results, Tateishi et al. described the selective cytotoxicity of the NAMPT inhibitors FK866 and GMX1778 in IDH1 mutant cancer cells [19]. They found that the downregulation of NAPRT, which is the rate-limiting enzyme of another NAD^+^ salvage pathway [36], causes a drop in NAD^+^ levels in IDH1^R132H^ cells and, thus, a susceptibility of those cells to NAMPT inhibition. Since we previously showed that the cell model we used lacked NAPRT, independent of the IDH1 status [17], this did not explain the observed effects in this study. Our data suggest that the cytotoxic effects of NAMPT inhibitors are, in part, independent of IDH1 status, thus possibly limiting their suitability as a selective treatment option for IDH1 mutated glioma. However, we note that this finding was based on NAMPT inhibitor treatment of different clones of one patient-derived glioblastoma cell line. Glioma arising from different cellular backgrounds may respond differently to NAMPT inhibition according to varying metabolic properties, such as variations in the expression of NAD^+^ synthesis enzymes [17]. Therefore, future studies are required to validate our findings regarding pharmacological NAMPT inhibition, ideally in patient-derived IDH mutant glioma models.

In conclusion, our data underline that targeting the NAMPT NAD^+^ regeneration pathway as a promising therapeutic option for gliomas with IDH1^R132H^ mutation. However, the method of treatment should be carefully considered since NAMPT inhibition with small molecules might not be effective, depending on the individual molecular background of a tumor. New efforts in therapeutic methods, such as the targeted delivery of siRNA [47,48] or the use of extracellular vesicles as drug delivery systems [49], may allow us to selectively reduce NAMPT expression in IDH1^R132H^ glioma and might be worth investigating in future studies.

## 4. Materials and Methods

### 4.1. Cell Culture and Reagents

HT7606 is a previously established primary glioblastoma cell line obtained from a patient who underwent surgery at the Klinik und Poliklinik für Neurochirurgie, University Hospital Carl Gustav Carus, TU Dresden, after informed written consent and with approval of the local ethics committee [31]. The cells and all derived cell lines were cultured in Dulbecco’s Modified Eagle Medium containing 4.5 g/L glucose, GlutaMAX™ and pyruvate, which was supplemented with a 10 mM Hepes Buffer, 4× Non-Essential Amino Acids, 100 U/mL penicillin/streptomycin (all from Gibco, Waltham, MA, USA) and 20% fetal bovine serum (Biochrom AG, Berlin, Germany). The cells were cultured in a humidified incubator at 37 °C containing 5% CO_2_.

### 4.2. Genome Editing Using CRISPR/Cas9

The IDH1 c.395G > A point mutation was introduced in the HT7606 cell line using CRISPR/Cas9. Two different protocols were applied in three independent experiments using either the Cas9-plasmid pX458 (#48138 Addgene, Watertown, MA, USA) according to the protocol of Ran et al. 2013 [50] or the Cas9-NLS-tagRFP (Eupheria Biotech, Dresden, Germany) following the manufacturer’s protocol. The cells were transfected using Lipofectamine 3000 (Thermo Fisher Scientific, Waltham, MA, USA). The target-specific gRNA 5′-GGGGATCAAGTAAGTCATGT-3′ was designed on the crispr.mit.edu platform. It was used for all experiments and either transfected after ligation into the pX458 plasmid or as RNA molecules. Additionally, a single-strand oligodeoxynucleotide was transfected as a DNA-repair template. Successfully transfected clones were selected using FACS and seeded as single cells. Edited clones were screened with allele-specific PCR using two different sets of primers. In the three experiments, 142, 243 and 88 clones were analyzed. In each of the experiments, the IDH1^R132H^ mutation was found in one clone, resulting in editing efficiencies of 0.4% to 1.1%. IDH1 c.395G > A point mutation validation and gRNA off-target screening were accomplished using Sanger sequencing (primers are listed in Appendix A, off-target regions are listed in Appendix A). IDH1^R132H^ expression was confirmed with cDNA Sanger sequencing and Western blotting.

### 4.3. Quantification of NAD^+^/NADH and NADP^+^/NADPH

NAD^+^, NADH, NADP^+^ and NADPH levels were quantified using the bioluminescent NAD/NADH-Glo™ Assay and the NADP/NADPH-Glo™ Assay (both Promega, Madison, WI, USA) following the manufacturer’s protocol for measuring NAD(P)^+^ and NAD(P)H individually. Per well, 8.000 cells were seeded in a 96-well plate. After 24 h, either the IDH1^R132H^ small molecule inhibitor AGI-5198 (1 µM; Merck Millipore GmbH, Darmstadt, Germany) or DMSO (0.001%) was added. After 48 h, cells were treated with 0.2 N NaOH with 1% dodecyltrimethylammonium bromide for lysis. Half of the lysed cell sample was then incubated for 15 min at 60 °C for the NAD(P)H measurement. The other half was treated with 0.4 N HCl and then incubated for 15 min at 60 °C for the NAD(P)^+^ measurement. The samples were again split and incubated with either the NAD/NADH-Glo Detection reagent or the NADP/NADPH-Glo Detection reagent at room temperature in the dark for 35 min. Luminescence was measured with a Mithras LB940 microplate reader. Calculation of the NAD^+^, NADH, NADP^+^ and NADPH concentrations was done with a standard curve using the corresponding metabolites.

### 4.4. RNA Extraction, Protein Extraction and Western Blotting

GeneMATRIX Universal DNA/RNA/Protein Purification Kit (Roboklon, Berlin, Germany) was used to extract RNA and proteins from the same samples following the manufacturer’s instructions. The concentration of RNA was determined with a Qubit^®^ RNA HS Assay Kit and a Qubit^®^ 2.0 Fluorometer (Life Technologies, Carlsbad, CA, USA). The extracted proteins were quantified using a Pierce™ BCA Protein Assay Kit (Pierce Biotechnology, Pittsburgh, PA, USA) and an Infinite M200 with Magellan™ Data Analysis Software (TECAN Group AG, Mannedorf, Switzerland). Proteins were separated on NuPAGE 4–12% Bis-Tris gels (Invitrogen, Waltham, MA USA) for 45 min at 200 V (15–30 µg protein per lane) and transferred onto Hybond ECL Membranes (GE Healthcare, Chicago, IL, USA) for 1 h at 30 V using a NuPAGE^®^ Transfer Buffer (Thermo Fisher Scientific, Waltham, MA, USA).

The membranes were washed twice with phosphate-buffered saline with 0.05% Tween 20 and blocked in 5% (*w*/*v*) nonfat milk for 60 min at room temperature before overnight (4 °C) incubation with primary antibodies against NAMPT (P4D5AT, Enzo Life Science, Inc., Farmingdale, NY, USA; 1:1500), IDH1 (ab117976, Abcam, Cambridge, UK; 1:1000), IDH1^R132H^ (DIA-H09, Dianova, Hamburg, Germany; 1:250), nicotinamide riboside kinase (NMRK1; PA5-26654, Thermo Fisher Scientific, Waltham, MA, USA; 1:1000), nicotinamide mononucleotide adenylyltransferase (NMNAT1; HPA059447, Sigma-Aldrich Corp., Burlington, MA, USA; 1:1000) and NAD^+^ kinase (NADK; HPA048909, Sigma-Aldrich Corp., Burlington, MA, USA; 1:500). Antibodies against glyceraldehyde 3-phosphate dehydrogenase (GAPDH; H86045M, Meridian Life Science, Memphis, TN, USA; 1:2,000,000) and Actin (A2228, Sigma-Aldrich Corp., Burlington, MA, USA; 1:1000) were used as a loading control. All antibodies were diluted in 5% (*w*/*v*) nonfat milk. Human recombinant NAMPT protein (Abnova, Taipei, Taiwan) served as a positive control. Subsequently, the membranes were probed with an anti-mouse secondary antibody (AP127P, Merck Millipore GmbH, Darmstadt, Germany; 1:10,000) for 1 h at room temperature. The Lumi-LightPLUS Western Blotting Substrate (Roche, Basel, Switzerland) was used for band detection with a Gel iX20 Imager (INTAS Science Imaging Instruments GmbH, Göttingen, Germany). The intensity of the Western blot bands was adjusted to the GAPDH control bands and quantified using the ImageJ freeware (http://rsb.info.nih.gov/ij/index.html, accessed on 25 March 2020).

### 4.5. Real-Time Quantitative PCR (RT-qPCR)

cDNA was synthesized from RNA extracts using the SuperScript™ VILO cDNA Synthesis Kit (Thermo Fisher Scientific, Waltham, MA, USA). RNA from normal brain tissue (Human adult normal tissue: Brain and Human Adult Normal Tissue 5 Donor Pool: Brain, both from Biochain Institute Inc., Newark, CA, USA) was used as the control. Quantitative analysis was performed with the SYBR™ Green PCR Master Mix (Applied Biosystems, Bedford, MA, USA) and a CFX96 Touch™ Real-Time PCR Detection System (Bio-Rad Laboratories, Hercules, CA, USA) according to the manufacturer’s instructions. Primers are listed in Appendix A. The relative gene expression was calculated via normalization to the reference genes GAPDH and ARF1 according to the comparative Ct method [51].

### 4.6. esiRNA Knockdown

Endoribonuclease-prepared small interfering RNA (esiRNA) targeting NAMPT (esiNAMPT-A Catalog No. HU-05878-1, esiNAMPT-B Catalog No. esiSEC), nicotinamide mononucleotide adenylyltransferase (esiNMNAT1, Catalog No. HU-01617-1), NMRK1 (esiNMRK1 Catalog No. HU-09786-1) and NAD^+^ kinase (esiNADK Catalog No. HU-08503-1) were purchased from Eupheria Biotech GmbH (Dresden, Germany). esiRNA targeting the sea pansy enzyme Renilla luciferase (mock control, Catalog No. RLUC) was used as a non-targeting control. Cells were transfected with esiRNA using Lipofectamine™ 3000 (Thermo Fisher Scientific, Waltham, MA, USA). The components were diluted in Opti-MEM™ Reduced Serum Medium (Gibco, Waltham, MA, USA). Proper concentrations were confirmed in standard transfection efficiency experiments. Knockdown efficiencies were determined by RT-qPCR and Western blot analysis.

### 4.7. NAMPT Inhibition Using Small-Molecule Inhibitors

For the NAMPT small-molecule inhibitor treatment, cells were seeded in 96-well plates. After 48 h, the cell culture medium was replaced with a medium containing different concentrations of one of the NAMPT small-molecule inhibitors FK866, GMX1778 (both from Sigma-Aldrich Corp., Burlington, MA, USA) and GNE-617 (ApexBio, Houston, TX, USA) which were previously diluted in DMSO (final concentration 0.001%).

### 4.8. WST-1 Assay

Cells were seeded with 2000 cells per well in transparent 96-well plates and 10 µL of WST-1 reagent (Sigma-Aldrich Corp., Burlington, MA, USA) were added to each well. After incubation for 4 h at 37 °C in 5% CO_2_, absorption was measured at 440 nm with 690 nm as a reference wavelength using an Infinite M200 with Magellan™ Data Analysis Software v.7.2 (TECAN Group AG, Mannedorf, Switzerland).

### 4.9. Combinatorial esiNAMPT and GMX1778 Treatment

Cells were seeded in Corning^®^ 384 well black/clear bottom plates with 500 cells per well. esiNAMPT-A was applied in reverse transfection with Lipofectamine™ RNAiMAX (Thermo Fisher Scientific, Waltham, MA, USA) on the day of cell seeding. GMX1778 was applied 48 h after the esiRNA treatment or 24 h after cell seeding for a GMX1778 single treatment. Cell viability was measured with automated picture analysis as described below.

### 4.10. Measurement of Cell Viability with CyQUANT^®^ Direct Cell Proliferation Assay

Cell viability was analyzed with the CyQUANT^®^ Direct Cell Proliferation Assay (Molecular Probes Inc., Thermo Fisher Scientific, Waltham, MA, USA) following the manufacturer’s instructions. Fluorescence was measured at an excitation wavelength of 480 nm and emission wavelength of 535 nm using an Infinite M200 with Magellan™ Data Analysis Software (TECAN Group AG, Mannedorf, Switzerland).

### 4.11. Measurement of Cell Viability with Automated Picture Analysis

A viability assay after the described treatments was performed in black 384-well plates (Greiner, Frickenhausen, Germany). Cells were stained with Hoechst 33,342 and propidium iodide (Invitrogen™, Waltham, MA, USA). Plates were incubated at 37 °C for 10 min and then analyzed on an Operetta High-Content Imaging System (PerkinElmer, Waltham, MA, USA). Pictures were evaluated using Harmony™ software (PerkinElmer, Waltham, MA, USA). Living cells were calculated by subtracting the number of propidium iodide stained cells from the number of Hoechst-33,342-stained cells in each well.

### 4.12. Statistics

Statistical analysis was performed using GraphPad Prism 5 (GraphPad Software, Inc., San Diego, CA, USA). All data are presented as mean ± SEM. The number of cell lines (biological replicates, nb) and technical replicates (nt) for specific experiments are indicated in the figure captions. Single groups were compared using an unpaired t-test. Multiple groups were analyzed with one-way analysis of variance (ANOVA) followed by Dunnett’s post hoc test. Any *p*-values < 0.05 were considered statistically significant.

## Figures and Tables

**Figure 1 ijms-23-05787-f001:**
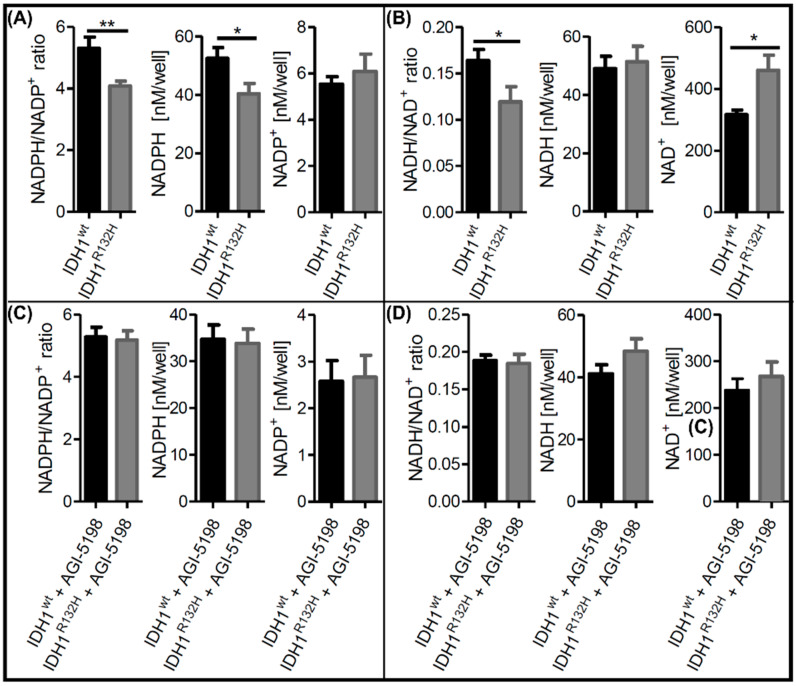
Alterations in NAD^+^ and NADP^+^ metabolism can be directly attributed to the neoenzymatic function of IDH1^R132H^. (**A**,**C**) Intracellular NADPH/NADP^+^ ratios, as well as NADPH and NADP^+^ levels, in untreated IDH1^wt^ and IDH1^R132H^ cells before (**A**) and after treatment with the selective mutant IDH1 inhibitor AGI-5198 for 48 h (**C**). (**B**,**D**) Intracellular NADH/NAD^+^ ratios, as well as NADH and NAD^+^ levels, in untreated IDH1^wt^ and IDH1^R132H^ cells before (**B**) and after treatment with the selective mutant IDH1 inhibitor AGI-5198 for 48 h (**D**) (nb = 3 per group; nt = 3; * *p* ≤ 0.05, ** *p* ≤ 0.01).

**Figure 2 ijms-23-05787-f002:**
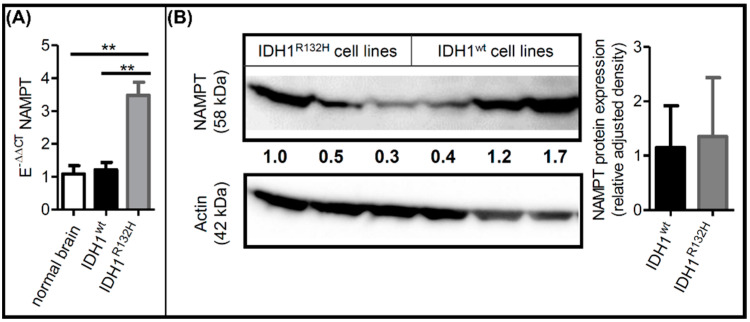
IDH1^R132H^ altered NAMPT expression. (**A**) Relative NAMPT mRNA levels in comparison to reference genes GAPDH and ARF1 (E^-ΔCT^). Relative NAMPT mRNA levels in IDH1^wt^ (nb = 2) and IDH1^R132H^ cells (nb = 3) were compared with the relative NAMPT mRNA levels in a normal brain (E^-ΔΔCT^) (nt = 3). (**B**) Western blot analyses of NAMPT in IDH1^wt^ cell lines (HT7606-IDH1^wt/wt−40^, HT7606-IDH1^wt/wt−141^, HT7606) and IDH1^R132H^ cell lines (HT7606-IDH1^R132H/wt−1^, HT7606-IDH1^R132H/wt−16^, HT7606-IDH1^R132H/wt−88^). Data are presented as the ratio of NAMPT to the reference protein GAPDH (nb = 3; nt = 2) (** *p* ≤ 0.01).

**Figure 3 ijms-23-05787-f003:**
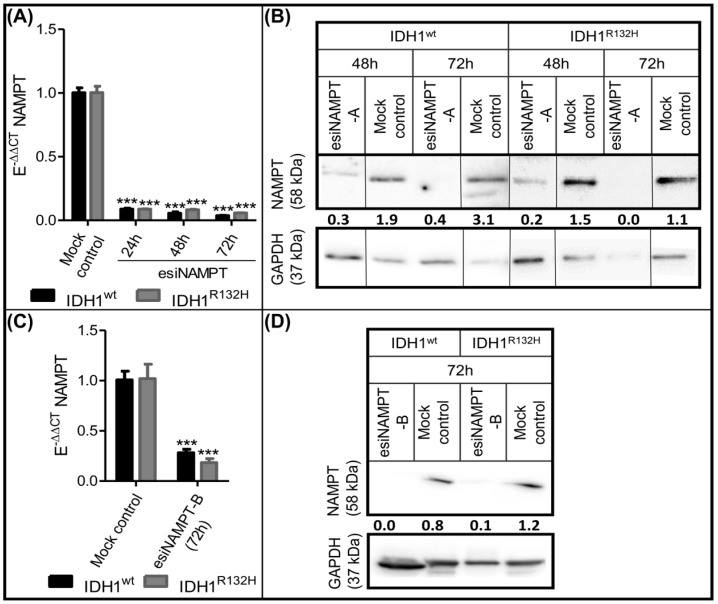
Successful knockdown of NAMPT. (**A**,**C**) Relative NAMPT mRNA levels compared with the reference genes GAPDH and ARF1 (E^-ΔCT^) in IDH1^wt^ and IDH1^R132H^ cells after treatment with esiNAMPT-A (**A**) or esiNAMPT-B (**C**) were compared with the relative NAMPT mRNA levels in the respective cells after treatment with the mock control (E^-ΔΔCT^). Data are presented as the ratio to treatment with the mock control. (nt = 3; *** *p* ≤ 0.001 compared with the mock control). (**B**,**D**) Western blot analyses of NAMPT in IDH1^wt^ (HT7606) and IDH1^R132H^ (HT7606-IDH1^R132H/wt−1^) cells after treatment with the mock control, esiNAMPT-A (**B**) or esiNAMPT-B (**D**) for 48 or 72 h; black lines indicate where bands from the same gel were ordered differently for improved clarity. Data are presented as the ratio of NAMPT to GAPDH.

**Figure 4 ijms-23-05787-f004:**
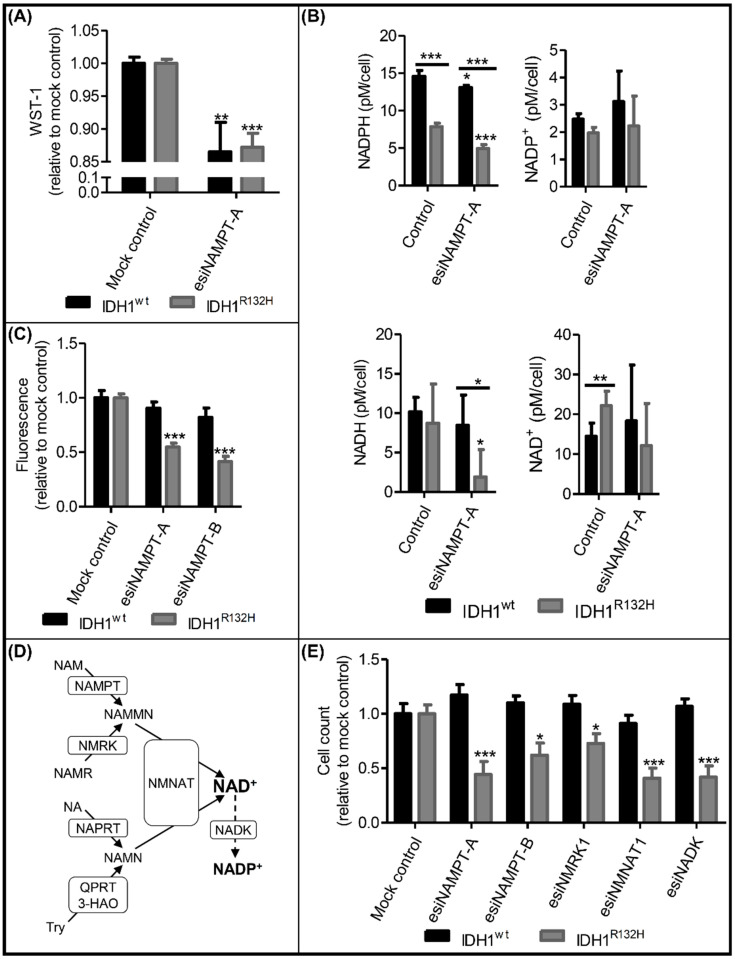
Knockdown of NAMPT and other NAD^+^ synthesis enzymes selectively reduced the viability of IDH1^R132H^ cells. (**A**) Metabolic activity in IDH1^wt^ and IDH1^R132H^ cells after treatment with esiNAMPT-A for 48 h (nb = 3). (**B**) NADPH, NADP^+^, NADH and NAD^+^ levels after 48 h of treatment with DMSO (control) or esiNAMPT-A. Each condition was normalized to the cell count of each sample (nb = 2 per group). (**C**) Cell viability after treatment with esiNAMPT-A or esiNAMPT-B for 72 h (nb = 2 for IDH1^wt^; nb = 3 for IDH1^R132H^). (**D**) NAD^+^ synthesis and salvage pathways. NAD^+^ is synthesized de novo from tryptophan (Try) or salvaged from nicotinamide (NAM), nicotinamide riboside (NAMR) or nicotinic acid (NA). NAD^+^ kinase (NADK) generates NADP^+^ from NAD^+^ and ATP. NAMPT: nicotinaminde phosphoribosyltransferase, NMRK: nicotinamide riboside kinase, NAPRT: nicotinic acid phosphoribosyltransferase, 3-HAO: quinolinic acid-synthesis-enzyme 3-hydroxyanthranilate 3,4-dioxygenase, QPRT: quinolinic acid phosphoribosyltransferase, NMNAT: nicotinamide mononucleotide adenylyltransferase, NAMMN: nicotinamide mononucleotide, NAMN: nicotinic acid mononucleotide. (**E**) Cell count after treatment of IDH1^wt^ and IDH1^R132H^ cells with esiNAMPT-A, esiNAMPT B, esiNMRK1, esiNMNAT or esiNADK for 72 h (nb = 3 per group). All data are presented as a ratio to treatment with the mock control/control (nt = 3; * *p* ≤ 0.05, ** *p* ≤ 0.01, *** *p* ≤ 0.001 compared with the mock control/control unless indicated otherwise).

**Figure 5 ijms-23-05787-f005:**
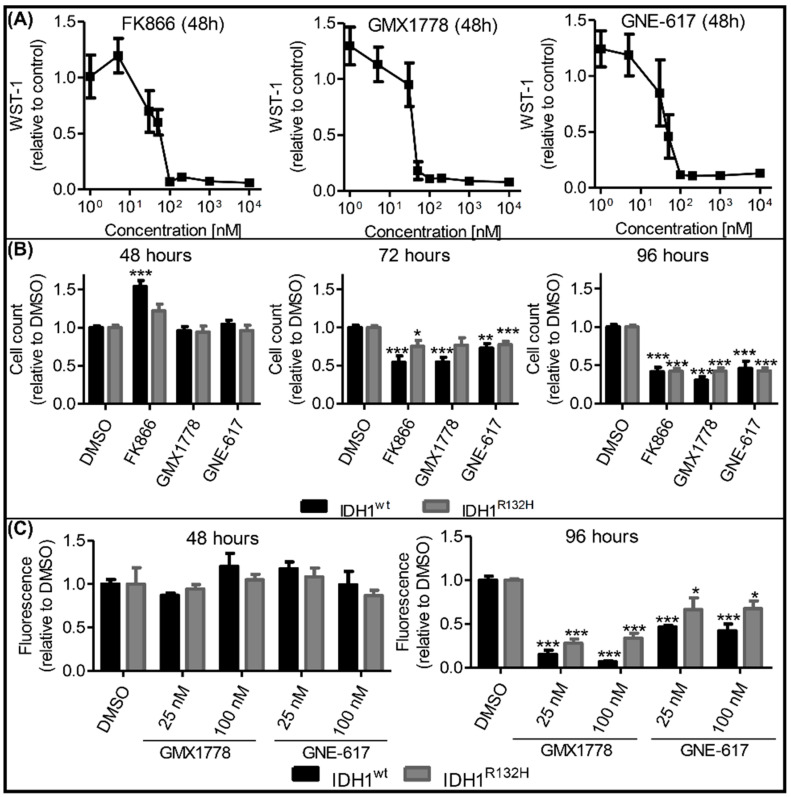
Effects of NAMPT inhibitors on cell viability were independent of IDH1 status. (**A**) Metabolic activity of IDH1^wt^ cells (HT7606) after treatment with different concentrations of the NAMPT inhibitors FK866, GMX1778 and GNE-617 for 48 h. Data are presented as a ratio to treatment with DMSO (nb = 3). (**B**) Cell count after treatment with 100 nM of FK866, GMX1778 and GNE-617 for 48, 72 or 96 h. Data are presented as a ratio to treatment with DMSO (nb = 3 per group). (**C**) Cell viability after treatment with 25 nM or 100 nM of GMX1778 or GNE-617 for 48 or 96 h. Data are presented as a ratio to treatment with DMSO (nb = 2 per group). Each condition was normalized to the cell count of each sample (nb = 2 per group) (nt = 3; * *p* ≤ 0.05, ** *p* ≤ 0.01, *** *p* ≤ 0.001 compared with DMSO unless indicated otherwise).

**Figure 6 ijms-23-05787-f006:**
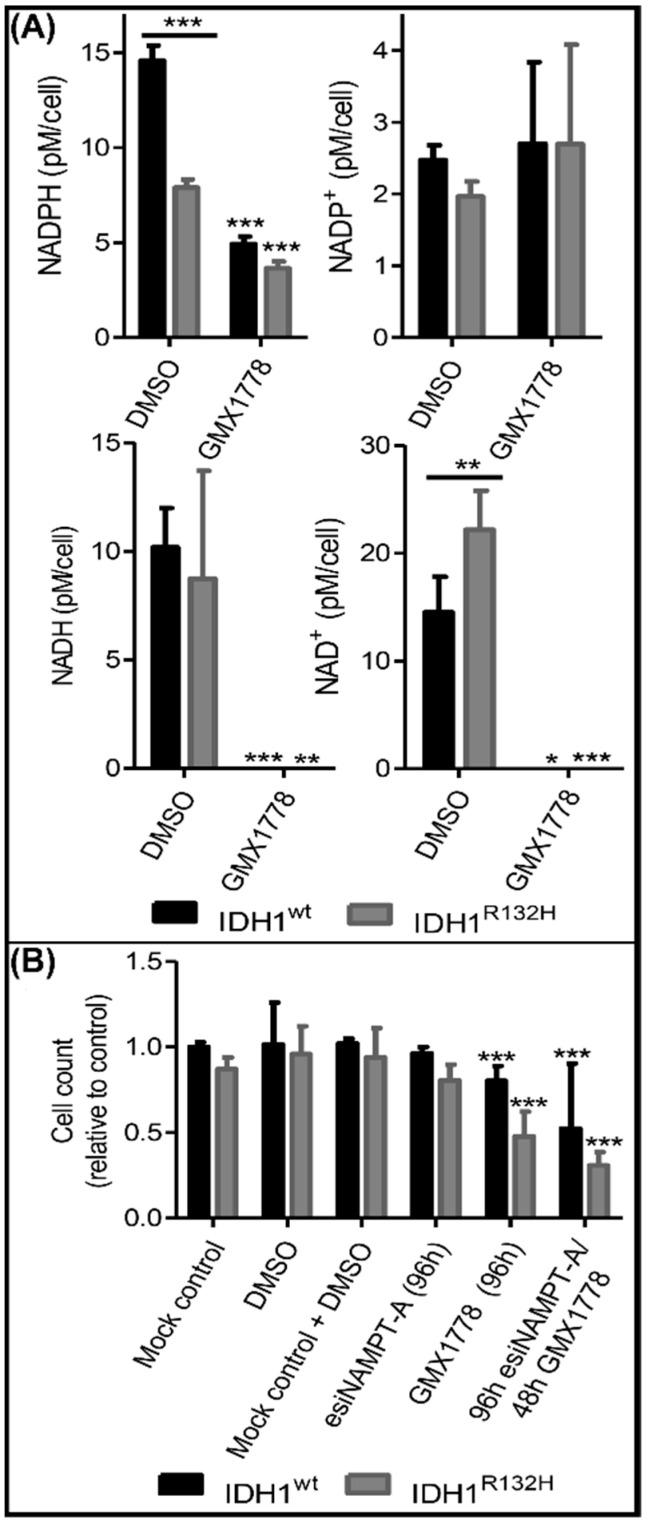
Combinational esiNAMPT knockdown and GMX1778 treatment. (**A**) NADH and NADPH levels after 48 h of treatment with DMSO or 25 nM GMX1778. Each condition was normalized to the cell count of each sample (nb = 2 per group) (* *p* ≤ 0.05, ** *p* ≤ 0.01, *** *p* ≤ 0.001 compared with DMSO unless indicated otherwise). (**B**) The cells were treated for 96 h with a mock control, DMSO, mock control and DMSO, esiNAMPT-A or 25 nM GMX1778. The esiNAMPT-A + GMX1778 treatment was executed by treating cells for 48 h with 25 nM GMX1778 after 48 h of esiNAMPT-A pretreatment (nb = 2 per group). All data present the ratio to control cells in a medium without any treatment (not shown) (nt = 3; * *p* ≤ 0.05, *** *p* ≤ 0.001 compared with the mock control).

## Data Availability

The data used for this study are contained within the article or Appendix A.

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
