# Peer review of "Different Effects of RNAi-Mediated Downregulation or Chemical Inhibition of NAMPT in an Isogenic IDH Mutant and Wild-Type Glioma Cell Model"

_ijms, 2022, doi:10.3390/ijms23105787_

Round 1
Reviewer 1 Report
The paper studies a clinically significant mutation in glioblastomas using in vitro models.
The methodology is sound and the paper presents very interesting and somewhat unexpected, but valuable results about the small molecule inhibitors.
The authors include detailed information on the methodology used, primers and images. Article is clear and well written and authors successfully compare to similar relevant literature.
Given the above, I recommend the article can be published in present form.
Author Response
We thank you very much for the review and the positive feedback on our manuscript.
Reviewer 2 Report
Figure 3b, the western blot figures have several vertical lines between lanes: authors should put a black color line in-between to clearly indicate this and explain in the figure legend that figures are assembled from different part of the same gel.
Author Response
Figure 3b, the western blot figures have several vertical lines between lanes: authors should put a black color line in-between to clearly indicate this and explain in the figure legend that figures are assembled from different part of the same gel.
We thank you very much for reviewing our manuscript. As suggested, we added black lines to figure 3b and amended the figure legend accordingly to indicate that the image was assembled from different parts of a gel.
Reviewer 3 Report
This manuscript deals with the generation of a heterozygous mutant for IDH1R132H using CRISPR/Cas procedure of the glioma cell line HT7606 and the beahviour in this cellular model was compared to the IDH1-wild-type (IDH1wt) cells.
The authors performed a detailed analysis of the nicotinamide phosphoribosyltransferase (NAMPT) activity on the metabolic activity and viability, and they used NAMPT inhibitors FK866, GMX1778 and GNE-617 to study their potential application for therapy against the model proposed. They found that these inhibitors have some off-target effects, and so they should be used with attention.
Although the data can be of interest and the experiments are well performed and presented, the data derived from the analysis of a single cell line and this is a very strong limitation of this study. I strongly suggest confirming the data on additional cell lines.
In the title should be mentioned that a cell line has been used.
Author Response
Although the data can be of interest and the experiments are well performed and presented, the data derived from the analysis of a single cell line and this is a very strong limitation of this study. I strongly suggest confirming the data on additional cell lines.
his study. I strongly suggest confirming the data on additional cell lines.
We thank you very much for reviewing our manuscript. As you pointed out, our experimental findings can be of interest for potential treatment strategies specifically targeting IDH-mutated gliomas. We are aware that, however, using one cell model (even when using different IDH-mutated and wt clones) presents a limitation of our study, since effects might differ in another genetic background. We addressed this point in more detail in the discussion to better clarify this limitation of this study and to point out the importance of confirming our results in future studies on additional cell models (see lines 448-464 / 493–512 showing tracked changes).
In the title should be mentioned that a cell line has been used.
We did change the title to clarify that the study was done using one cell line to: Different effects of RNAi-mediated downregulation or chemical inhibition of NAMPT in an isogenic IDH mutant and wild-type glioma cell model.
Round 2
Reviewer 3 Report
The authors have revised the manuscript following the concerns raised.